# Psychiatric symptomatology in skin-restricted lupus patients without axis I psychiatric disorders: A post-hoc analysis

**Fabien Rondepierre**[1], **Urbain Tauveron-Jalenques**[2], **Solène Valette**[2], **Aurélien Mulliez**[3], **Michel D'Incan**[4], **Sophie Lauron**[2], **Isabelle Jalenques**[2]*

**1** CHU Clermont-Ferrand, Service de Psychiatrie de l'Adulte A et Psychologie Médicale, Clermont-Ferrand, France, **2** Clermont Auvergne Université, CNRS, CHU Clermont-Ferrand, Service de Psychiatrie de l'Adulte A et Psychologie Médicale, Clermont Auvergne INP, Institut Pascal, Clermont-Ferrand, France, **3** CHU Clermont-Ferrand, Direction de la Recherche Clinique et de l'Innovation, Clermont-Ferrand, France, **4** Clermont Auvergne Université, INSERM, CHU Clermont-Ferrand, Service de Dermatologie, Clermont Auvergne INP, Institut Pascal, Clermont-Ferrand, France

* ijalenques@chu-clermontferrand.fr

## Abstract

### Background

Skin-restricted lupus is a chronic inflammatory disease associated with high rates of depression and anxiety disorders. Patients without psychiatric disorders can experience anxiety and depressive symptoms at a subclinical level, which could be risk factors for progression towards psychiatric disorders. It was decided, therefore, to investigate the presence of specific symptoms in skin-restricted lupus patients without axis I psychiatric disorders and their impact on the occurrence of axis I psychiatric disorders during the study follow-up.

### Methods

Longitudinal data of 38 patients and 76 matched controls without active axis I psychiatric disorders from the LuPsy cohort were used. Depressive, neurovegetative, psychic and somatic anxiety symptom scores were established from the Montgomery-Asberg Depression Rating Scale (MADRS) and the Hamilton Anxiety Rating scale (HAMA).

### Results

None of the participants had any current active axis I psychiatric disorders but the patients had personality disorders more frequently and had received more past psychotropic treatments than the controls. They also had higher MADRS and HAMA scores than the controls, in particular neurovegetative, psychic anxiety and somatic symptoms scores. No dermatological factor tested was associated with these scores, whereas being a lupus patient was associated with higher neurovegetative and somatic symptoms scores, having a current personality disorder with higher depressive and neurovegetative scores and receiving more past psychotropic treatments with psychic anxiety and somatic symptoms scores. The occurrence of psychiatric disorders during the study follow-up was associated with an

**Data Availability Statement:** All the data are now available at Mendeley: Rondepierre, Fabien; Tauveron-Jalenques, Urbain; Valette, Solène;

Mulliez, Aurélien; D'Incan, Michel; Lauron, Sophie; Jalenques, Isabelle (2022), "Psychiatric symptomatology in skin-restricted lupus patients without psychiatric disorders: a post-hoc analysis", Mendeley Data, V1, doi: 10.17632/bdxf94z5xw.1.

**Funding:** This study was supported by a grant from the French Ministry of Health (PHRC IR 2006 Jalenques, N°2008-A00343-52) and from Société Française de Dermatologie, 2010 (D'Incan). The funders had no role in study design, data collection and analysis, decision to publish, or preparation of the manuscript.

**Competing interests:** The authors have declared that no competing interests exist.

elevated psychic anxiety score at baseline and past psychotropic treatment but not with history of psychiatric disorder.

## Limitations

The LuPsy cohort included a large number of patients with axis I psychiatric disorders, the sample without axis I psychiatric disorders is therefore limited.

## Conclusions

We observed numerous psychiatric symptoms among the skin-restricted lupus patients. They should therefore receive special attention in the management of their subclinical symptoms before they progress towards full psychiatric disorders.

## Introduction

Skin-restricted lupus (SRL) is a chronic inflammatory disease characterized by many elevated cytokines [1]. SRL includes discoid lupus erythematosus (DLE), lupus tumidus (LT) and subacute cutaneous lupus erythematosus (SCLE). It is associated with high rates of depression and anxiety disorders [2, 3] as in other chronic inflammatory diseases such as psoriasis, alopecia areata, atopic dermatitis, systemic lupus erythematosus, inflammatory bowel disease, multiple sclerosis and rheumatoid arthritis [4–9]. All these diseases are characterized by longstanding systemic inflammation that, in vulnerable patients, can cause neuroinflammation and lead to depression and anxiety [10, 11]. Individuals can be considered vulnerable on the basis of genetic predisposition, psychiatric history or an exacerbated immune response [12]. In this inflammatory context, other patients can develop anxious and depressive symptoms that are not severe enough to meet DSM criteria for depression and anxiety disorders but which correspond to subclinical disorders [13, 14]. These subclinical disorders could be risk factors for progression towards genuine depression and anxiety disorders [13, 15]. It would therefore be useful to identify specific symptoms associated with chronic inflammatory diseases. Additionally, interventions in preventing progression may be of interest [16].

The first mention of the depressive and anxious effects of inflammation were made in a study of interferon therapy [17]. A later study described the chronology of symptom onset in interferon-treated patients, who first developed neurovegetative and somatic symptoms and then anxious and depressive symptoms [18]. Since then, many studies have looked for links between inflammation and depressive symptoms, including in chronic inflammatory diseases [4, 6, 8, 9, 19].

Additionally, inflammation and depressive symptoms were observed in association in obese men and in diabetic patients [20, 21]. In acute coronary heart disease patients with no history of depression, inflammation was associated with somatic symptoms [22]. In the same study, a longitudinal association was also found between inflammation and anxious symptoms [22].

In the present exploratory study, we used longitudinal data to investigate the presence of depressive, psychic anxiety, neurovegetative and somatic symptoms in SRL patients without current active axis I psychiatric disorders. First, we compared baseline symptom scores in patients and controls, looking for associated factors, and second, we attempted to identify risk factors in patients who developed psychiatric disorders during follow-up.

## Methods

Data from the studies of the LuPsy cohort were used for this analysis. This cohort comprised 80 consecutive outpatients with chronic SRL (DLE and LT) or SCLE and 160 healthy control subjects without a history of lupus recruited among volunteers from a clinical research centre and workers in public hospitals, the national railway company and administrative personnel of the state education system. As reported elsewhere, the LuPsy studies were performed to compare the baseline prevalence of psychiatric and personality disorders in SRL patients and sex-, age-, and education level-matched controls [3, 23]. The cohort was approved by the local ethics committee (Comité de Protection des Personnes Sud-Est 6, reference 2008-A00343-52 / AU740, 18 June 2009). After presentation of the objectives and procedures of the study, all participants provided their written informed consent.

### Participants

As the objective of this study was to investigate the presence of psychiatric symptoms in SRL patients without active axis I psychiatric disorders, all patients of the LuPsy cohort with current axis I psychiatric disorders according to the Mini International Neuropsychiatric Interview (MINI 5.0.0) [24, 25] at baseline were excluded from the study, leaving 38 patients. They were matched for gender, age (plus or minus 5 years) and level of education with 76 healthy controls without active axis I psychiatric disorders from the LuPsy cohort. The patients were followed for 2 years with a visit every 6 months (5 visits in total) including both a dermatological and a psychiatric evaluation. Controls had no follow-up.

### Dermatological evaluation

The dermatologist determined the location and number of the lesions and the Cutaneous Lupus Erythematosus Disease Area and Severity Index score (CLASI) [26], whenever possible.

The date of the first symptoms of the SRL, the consumption of tobacco and the past and current specific treatments were collected.

### Psychiatric evaluation

The past and current axis I psychiatric disorders were explored by a psychiatrist using a structured interview, the MINI 5.0.0 [25]. The 99-item self-report Personality Diagnostic Questionnaire 4+ (PDQ-4+) was used to assess personality disorders. In its latest version, the clinical significance scale of the PDQ-4+ (CSS) which is an individual directive interview, allowed the psychiatrist to confirm or not the specific diagnosis of personality disorder suggested by a required number of positive criteria [27–29]. The psychiatric assessments could not be performed without perceiving the case-control status since most patients presented with skin lesions on a visible area.

### Symptom scoring

The psychiatrist also recorded scores from the Montgomery-Asberg Depression Rating Scale (MADRS) [30] and the Hamilton Anxiety Rating scale (HAMA) [31]. Depressive, neurovegetative, psychic anxiety and somatic symptom scores were calculated by adding the scores of the following selected items: (1) apparent sadness, reported sadness, inability to feel, pessimistic thoughts and suicidal thoughts from the MADRS for depressive symptoms; (2) reduced sleep, reduced appetite and lassitude from the MADRS for neurovegetative symptoms; (3) anxious mood, tension, fears and behavior at interview from the HAMA scale for psychic anxiety

symptoms and (4) muscular, sensory, cardiovascular, respiratory, gastrointestinal and genito-urinary symptoms also from the HAMA scale for somatic symptoms.

## Data management

Study data were collected and managed with Research Electronic Data Capture (REDCap) tools hosted at Clermont-Ferrand University Hospital [32]. The raw data supporting the conclusions of this manuscript are available at Mendeley [33].

## Statistical analysis

Statistics were computed with STATA V15 (Stata Corp, College Station, Texas, USA).

The study sample was described by frequencies and percentages for categorical data and by means ± standard deviations (or median and interquartile range when data not normal) for continuous data. Normality was assessed graphically and using Shapiro Wilk's test. Patients and controls were compared by Student's t-test (or Mann & Whitney test when data not normal) for continuous data and by chi-squared test (or Fisher's exact test when appropriate) for categorical data.

Depressive, neurovegetative, psychic anxiety and somatic symptom scores were compared in all participants according to sociodemographic and psychiatric factors and in patients according to dermatological factors by Mann & Whitney test for categorical data and by Spearman correlation coefficient for continuous data, in order to determine the associated factors.

Multivariate analyses of factors associated with the symptom scores were performed by multiple generalized linear regression, taking the group (patients vs controls) as main covariate and selecting other covariates, based on clinically relevant factors or statistically significant factors shown in univariate analysis. Results are shown as regression coefficient estimates and their 95% Confidence Interval (CI).

Those analyses were performed on the subgroup of patients using the same methods, to compare patients with and without occurrence of a psychiatric disorder during the follow-up. These analyses were completed by a factor analysis on mixed data using the dudi.mix function of the ade4 package in R (http://cran.r-project.org/web/packages/ade4/index.html) to visualize difference in patient's baseline profile according to the occurrence or not of a psychiatric disorder during the follow-up. The parameters considered in this analysis were those that could impact the development of psychiatric disorder and were determined based on the univariate results, author's experience and clinical relevance among sociodemographic (age, gender, smoking status), dermatological (number and size of lesion, type of lupus, medication) and psychiatric data (history, medication, consultation, symptom scores). Quantitative data were centered and scaled and qualitative data were converted into binary variables.

All tests were two-sided and a p-value <0.05 was considered statistically significant.

## Results

### Participant characteristics

The characteristics of the participants without current active axis I psychiatric disorders are similar to the whole LuPsy cohort and are given in Table 1. Briefly, patients were predominantly women (82%) and smokers (54%) with a mean age of 46.0 ± 15.4 years and median disease duration of 6.2 years [3.8–10.8]. Twenty eight (74%) patients had chronic lupus and 12 (26%) a subacute form. Twenty five patients (68%) had lesions on a visible area of the body. Thirty three patients (87%) were receiving treatment for lupus and 7 (18%) a psychotropic medication.

**Table 1. Participant characteristics.**

| | Patients | Controls | pValue |
|---|---|---|---|
| | N = 38 | N = 76 | |
| Sex (Female) | 31 (82) | 62 (82) | 1.000 |
| Age | 46.0 ± 15.4 | 46.5 ± 15.3 | 0.853 |
| Smokers | **50 (54)** | **15 (19)** | **< 0.001** |
| Medical comorbidities[†] | 17 (45) | 35 (46) | 0.894 |
| Lupus type | | | |
| Chronic cutaneous | 28 (74) | | |
| Subacute | 12 (26) | | |
| Lupus duration, years | 6.2 [3.8–10.8] | | |
| Largest lesion size, cm$^2$ | 1.5 [0.2–5.5] | | |
| Number of lupus lesions | 2.0 [1.0–4.0] | | |
| Number of affected areas | 1.0 [1.0–2.5] | | |
| Pruritus or burning sensations | 17 (46) | | |
| CLASI[††] | | | |
| Activity | 3.0 [0.5–4.0] | | |
| Damage | 1.0 [0.0–2.0] | | |
| Lesions on visible areas[†††] | 25 (68) | | |
| Current lupus treatment | 33 (87) | | |
| Synthetic antimalarials | 14 (37) | | |
| Thalidomide | 12 (32) | | |
| Topical steroids | 7 (18) | | |
| Current psychotropic treatment | 7 (18) | | |
| Antidepressant | 5 (13) | | |
| Anxiolytic | 4 (11) | | |
| Hypnotic | 1 (3) | | |
| Current psychiatrist consultation | 0 | | |
| Current personality disorder [††††] | **11 (32)** | **11 (14)** | **0.030** |
| Cluster A | 4 (11) | 5 (7) | 0.472 |
| Cluster B | 0 | 3 (4) | 0.550 |
| Cluster C | 6 (16) | 8 (11) | 0.381 |
| Depressive | 2 (5) | 3 (4) | 0.746 |
| Passive-aggressive (Negativistic) | 0 | 0 | |
| Past psychiatric disorder | 19 (50) | 29 (38) | 0.227 |
| Past psychotropic treatment | **19 (50)** | **18 (24)** | **0.005** |
| Antidepressant | 9 (24) | 10 (13) | 0.155 |
| Anxiolytic | **14 (37)** | **11 (14)** | **0.007** |
| Hypnotic | **6 (16)** | **2 (3)** | **0.016** |
| Past psychiatrist consultation | 6 (16) | 12 (16) | 1.000 |
| Baseline MADRS score | **3 [1–8]** | **1 [0–3]** | **0.005** |
| Baseline HAMA score | **7 [2–12]** | **3 [1–5]** | **0.001** |

Results are presented as N (%), mean ± SD or median [inter quartile range].

Bold values indicate significant results with a p-value < 0.05.

[†] The main comorbidities are cardiovascular diseases, which affect 19 participants (8 patients and 11 controls), digestive diseases (4 patients and 8 controls), musculoskeletal, and connective tissue diseases (5 patients and 6 controls) and tumors (3 patients and 7 controls).

[††] Value for 26 patients

[†††] A total of 23 areas were considered: face, neck, collar, ears, scalp, anterior and posterior sides of the chest, abdomen, lumbar area, left and right arms, left and right forearms, palmar sides of left and right hands, dorsal sides of left and right hands, left and right thighs, left and right legs, dorsal sides of left and right feet.

[††††] 14 missing data

The patients included more commonly smokers (54% vs 19%, p<0.001), had more frequently personality disorders (32% vs 14%, p = 0.030) and had received more past psychotropic treatment (50% vs 24%, p = 0.005) than controls. Unlike the controls, none of the patients, even among those who had a psychotropic treatment, had recently consulted a psychiatrist. Finally, patients had no more other medical comorbidities.

## MADRS, HAMA and symptom scores

None of the participants had any current active axis I psychiatric disorders but the patients recorded higher MADRS and HAMA scores than controls, in particular for neurovegetative, psychic anxiety and somatic symptoms (Fig 1 and Table 1). In contrast, the difference in the depressive symptom scores between patients and controls was non-significant (p = 0.092, Fig 1). The highest total MADRS and HAMA scores for patients were 19 and 24, respectively.

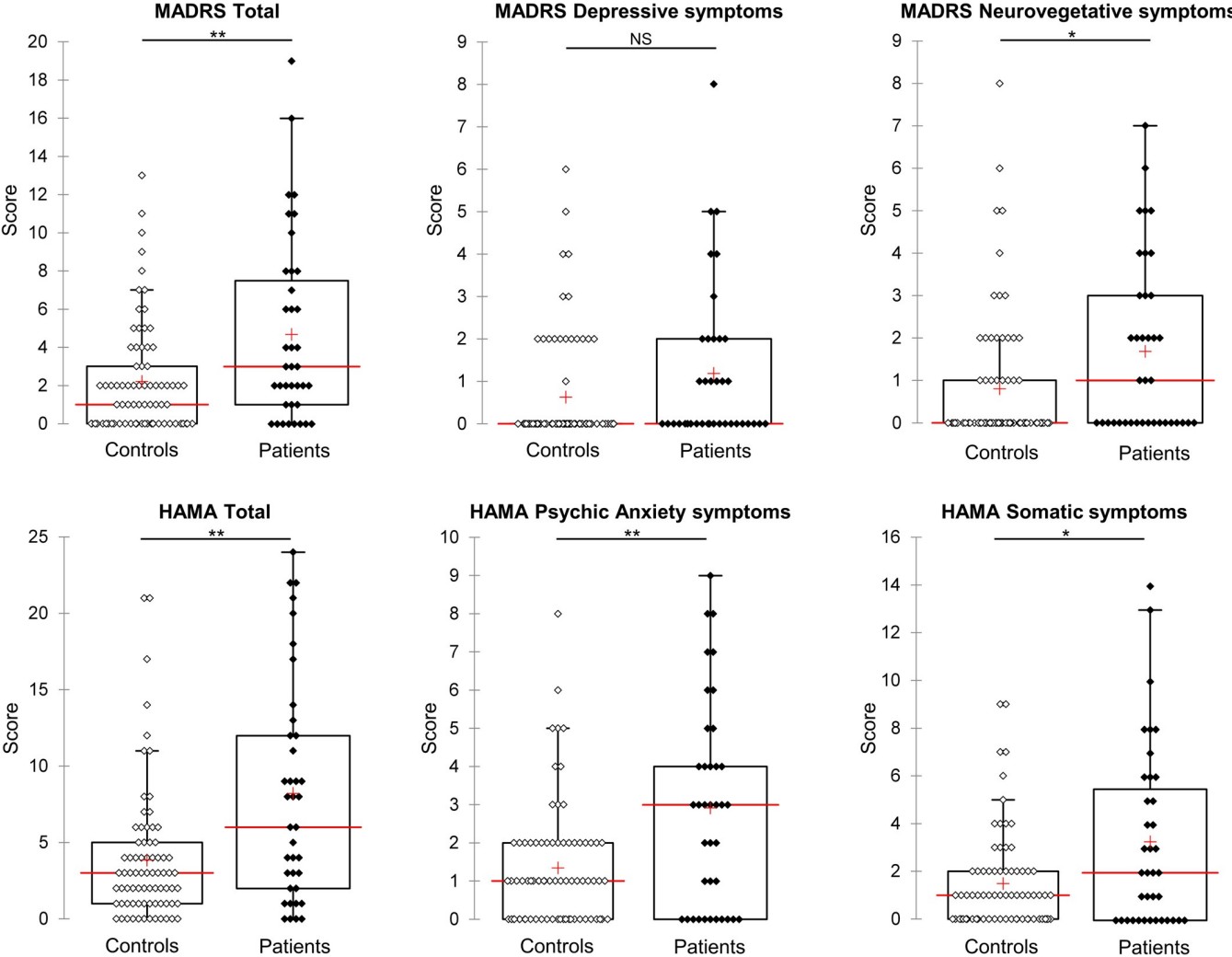

**Fig 1. Comparison of MADRS, HAMA and symptom scores between patients and controls.** Patients (in black) had significantly higher scores than controls (in white) for MADRS and HAMA total scores and for neurovegetative, psychic anxiety and somatic symptom scores. Scores are presented as Box plot with interquartile range and 95% confidence interval. Median and mean scores are presented as red line and red cross respectively. *p < 0.05 and ** p < 0.01. NS, non-significant.

## Factors associated with symptom scores in all participants

Each symptom score was compared according to sociodemographic and psychiatric character-istics to identify associated factors. The main factors identified were current personality disor-ders and past psychotropic treatment: participants with current personality disorders had higher depressive and psychic anxiety symptom scores than participants without (2 [0–4] vs 0 [0–0], p<0.001 and 2.5 [1–5] vs 1 [0–2], p = 0.009, respectively) and participants who had received psychotropic treatment had higher psychic anxiety and somatic symptom scores (2 [1–4] vs 1 [0–2], p = 0.001 and 2 [1–6] vs 1 [0–2], p = 0.004, respectively). Remarkably, no tested factor modified the neurovegetative symptom score (S1 Table).

To understand why patients had higher scores than controls and to identify the factors responsible for the differences, we performed multiple generalized linear regression using the study population (patients vs controls) and the previously identified factors (current personal-ity disorders and past psychotropic treatment) as covariates. The results are presented in Table 2. The depressive symptom score was related to the presence of a current personality dis-order whereas the neurovegetative symptom score was affected only by patient status and thus no other factor affected the score. Several factors affected the psychic anxiety, including being a SRL patient, having received psychotropic treatment in the past and having a current person-ality disorder. The somatic symptoms score was only associated with having received past psy-chotropic treatment.

## Factors associated with symptoms scores in patients

Because being an SRL patient was related to neurovegetative, psychic anxiety and somatic symptoms scores, we investigated whether dermatological factors affected these scores. We found no such association (S2 Table). The same psychiatric factors, mentioned above, were related to patients' scores (personality disorder, current antidepressant and past psychotropic treatments). Among the patients, women had higher scores for neurovegetative and somatic symptoms (S2 Table).

These identified factors were included in a multiple generalized linear regression to assess their impact on symptoms scores. The results confirmed that the depressive symp-toms score was only related to the presence of a current personality disorder (Coefficient estimates (CE) [95% CI], 2.05 [0.75; 3.35], p = 0.003). Interestingly, female patients had a higher neurovegetative symptom score (CE 1.99 [0.21; 3.77], p = 0.030), and patients with current antidepressant medication had a high psychic anxiety symptom score (CE 3.49 [0.43; 6.55], p = 0.027). In contrast, no factor in this model significantly modified the somatic score in SRL patients.

**Table 2. Associated factors with symptom scores in all participants.**

|  | Depressive symptoms | Neurovegetative symptoms | Psychic anxiety symptoms | Somatic symptoms |
|---|---|---|---|---|
| **Patients** | 0.35 [-0.28; 0.97] | 1.03 [0.24; 1.81]* | 0.97 [0.12; 1.82]* | 1.01 [-0.17; 2.18] |
| **Current personality disorder** | 1.46 [0.77; 2.14]*** | 0.18 [-0.69; 1.04] | 1.06 [0.13; 1.99]* | 0.55 [-0.74; 1.85] |
| **Current psychotropic treatment** | -0.31 [-1.65; 1.04] | -0.14 [-1.83; 1.55] | 1.67 [-0.16; 3.50] | 1.17 [-1.37; 3.71] |
| **Past psychotropic treatment** | 0.42 [-0.17; 1.02] | 0.18 [-0.57; 0.93] | 0.94 [0.12; 1.785]* | 1.64 [0.51; 2.76]** |

Multiple generalized linear regression provided coefficient estimates (CE) and 95% confident interval [95% CI]. Results are presented as CE [95% CI].

* p<0.05

** p<0.01

*** p<0.001

## Psychiatric evolution and associated baseline factors

Longitudinal data were unavailable for 5 of the patients; the other 33 underwent 4.2 assessments on average, with an average follow-up of 21.9 ± 5.9 months. Twenty-five (76%) patients completed the last assessment at 24 months. The distribution of patients' MADRS and HAMA scores was stable during follow-up (S1 Fig). Nine patients (27%) had a depressive or anxious disorder during follow-up: three depressive disorders (including one with anxiety disorder), two dysthymia, one suicide risk, two panic disorders and one social phobia. The psychiatric diagnosis was made at 12.8 ± 6.4 months of follow-up, at which time they were 47.0 ± 18.1 years old with lupus disease duration of 7.1 ± 4.8 years. The Fig 2A showed the correlation circle of the variables used in the factor analysis on mixed data. The first axis, which represented 24.7% of the variance of the data considered in the analysis, was more linked to the sex, HAMA and MADRS scores, whereas the second axis, with 15.3% of the variance, were more associated with the presence of past psychiatric disorder, of lupus or psychiatric medications. The Fig 2B revealed that the patients who developed a psychiatric disorder during the follow-up had a distinctive baseline profile than those who did not develop such disorders. These two profiles were mainly marked by different HAMA scores, with higher scores for the patients who developed a psychiatric disorder during the follow-up. Comparison of patients according to the occurrence or not of a psychiatric disorder confirmed that patients who developed a psychiatric disorder had higher psychic anxiety symptom scores at baseline (S3 Table). Although they had higher scores overall the only significant increases were in the psychic anxiety symptom scores (4 [3–6] vs 1.5 [0–3.5] p = 0.008) and the total HAMA scores (18 [6–21] vs 4.5 [1–10], p = 0.012) (Fig 3 and S3 Table). These patients had also received more past psychotropic treatment (hypnotic) (S3 Table). Four (44%) of the 9 had a psychiatric history but did not differ from those who had not developed psychiatric disorders during follow-up (S3 Table).

## Discussion

This cohort study indicated that SRL patients have a high prevalence of anxious and depressive symptoms. We show for the first time that even patients without current active axis I psychiatric disorders experience these symptoms and that over a 2-year follow-up period psychic anxiety symptoms at baseline were associated with the occurrence of psychiatric disorders.

Although the SRL patients were not suffering from established axis I psychiatric disorders they had higher depression (MADRS) and anxiety (HAMA) scores than the controls. Some of the scores were very high, in certain cases close to those of patients with confirmed depressive or anxious disorders, which is evidence of subclinical psychiatric symptoms. The scores were significantly higher for somatic, neurovegetative and psychic anxiety symptoms but only tended towards significance for depressive symptoms. The depressive and psychic anxiety symptoms scores were associated with personality disorders, which was unsurprising given the strong comorbidity of the disorders [34, 35]. However, in the patient group, depressive symptoms were associated only with personality disorders. Thus, the non-significant increase in depressive symptoms scores of the patients can be explained by a greater occurrence of personality disorders. For the other symptoms, various factors could be involved in the difference in scores between the patients and controls.

The psychic anxiety symptoms score was also related to past psychotropic treatment, as was the somatic symptoms score, but not to previous psychiatric disorders. Indeed, although the patients had a larger past consumption of psychotropic drugs they did not have a greater number of previous psychiatric disorders. In contrast, few patients, in comparison to controls, had consulted a psychiatrist. This could explain why patients who had no psychiatric disorders but only subclinical, mainly anxiety and somatic, symptoms, were prescribed psychotropic drugs.

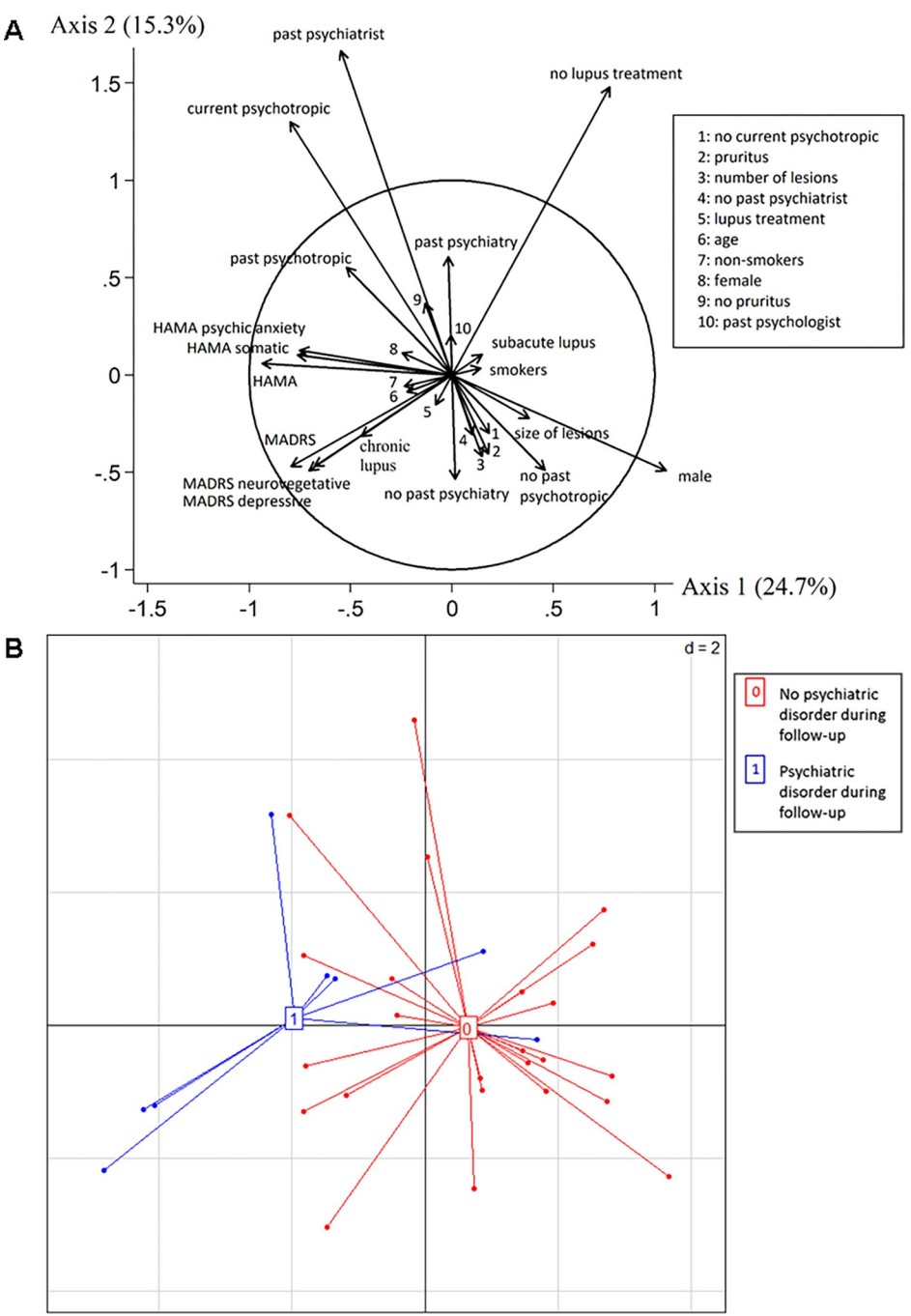

**Fig 2. Baseline profile of patients who developed a psychiatric disorder during the 2-years follow-up.** A: Correlation circles obtained with factor analysis showed the correlation between the variables. The shorter vectors represented variables with weak correlations and less contribution to the axis (axis 1 = 24.7% of the variance and axis 2 = 15.3%). B: Patients representation (one point for each patient) showing 2 clusters: one with psychiatric disorder during follow-up (1, blue) and another without psychiatric disorder (0, red). The variables most involved in these two clusters were all HAMA scores.

This result was confirmed in patients whose psychic anxiety symptoms score was associated with antidepressant treatment undergone during the study period. Interestingly, while most controls who were taking psychotropic drugs had consulted a psychiatrist no patient had.

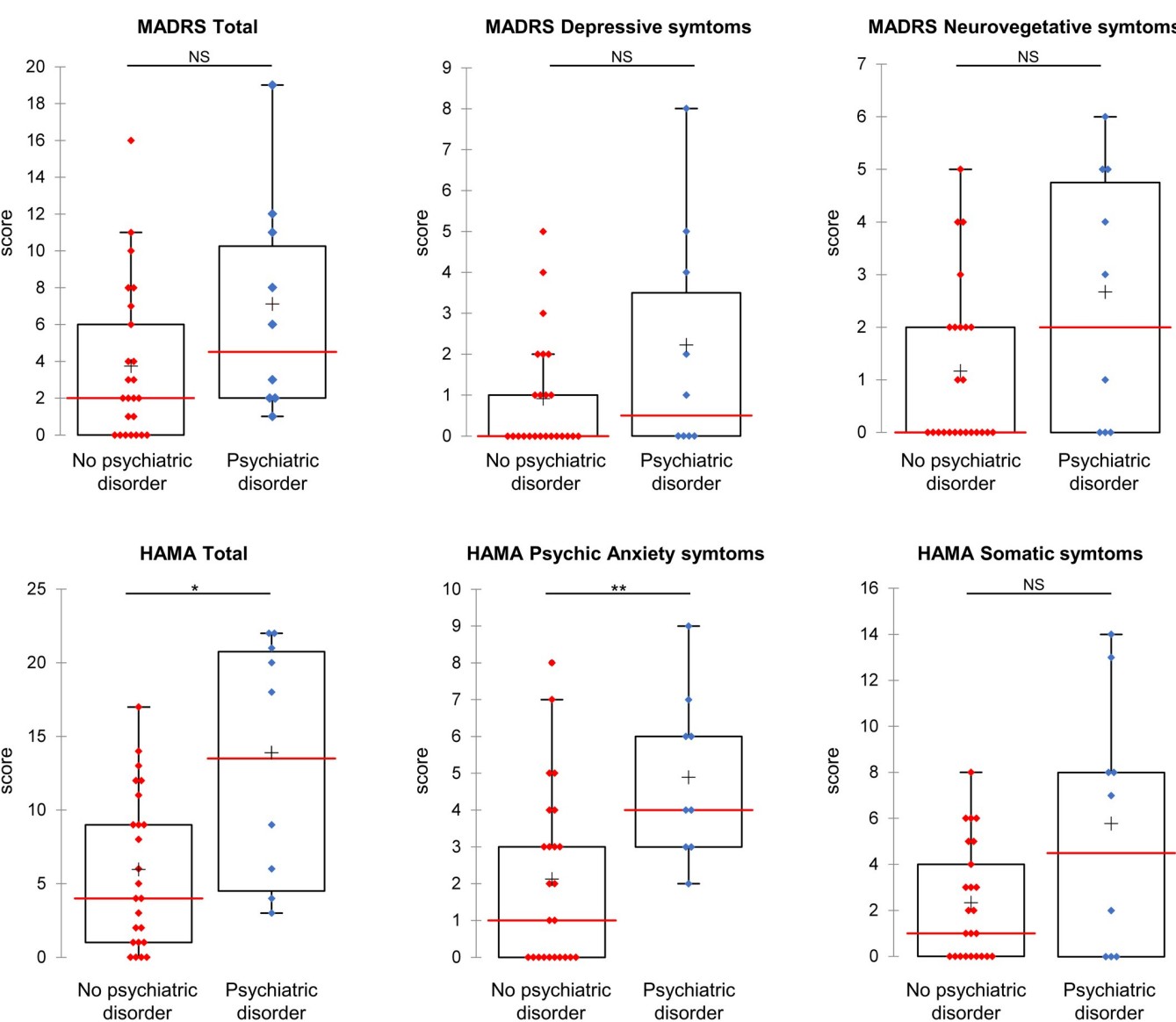

**Fig 3. Comparison of baseline MADRS, HAMA and symptom scores in patients according to the occurrence or not of a psychiatric disorder.** Patients who developed a psychiatric disorder are presented in blue (labelled psychiatric disorder), patients who never developed a psychiatric disorder during the follow-up are presented in red (labelled no psychiatric disorder). Baseline scores are presented as Box plot with interquartile range and 95% confidence interval. Median and mean scores are presented as red line and black cross respectively. Significant different: $^{*}$p < 0.05 and $^{**}$ p < 0.01.

These findings are consistent with those for the general population in France, which show widespread psychotropic drug use in individuals with no established psychiatry disorders [36]. The search for past psychiatric disorders was carried out through a psychiatric interview using the MINI, a method which is considered the gold standard. However, we cannot exclude defects in the recall of the disorders, or the psychotropic treatments or the psychiatrist consultations.

Being an SRL patient was related to high neurovegetative and psychic anxiety symptoms scores, irrespective of the other factors tested. Of these, the only significant factor was the neurovegetative score. These results suggest that SRL plays a role in the occurrence of these symptoms, possibly via chronic inflammation since no other dermatological factor had any effect. Several studies reported that all patients on interferon-α therapy had neurovegetative

symptoms [37, 38] and others that these symptoms were correlated with the level of inflammation [39]. In the latter report, Jokela et al. point to the correlation between C-reactive protein (CRP) and sleep problems, tiredness and changes in appetite, the specific neurovegetative symptoms of depression. Other studies on inflammation evidenced an increase in anxious and somatic symptoms [40, 41]. Unfortunately, we are unable to confirm with certainty the precise effect of inflammation on these symptoms because biological data are not available. However, none of the dermatological factors tested were associated with the different symptoms and thus the hypothesis of inflammation could be envisaged. A link between inflammation and psychiatric symptoms has already been established in other chronic inflammatory diseases such as SLE [42], psoriasis [43], ulcerative colitis [44], diabetes [20] and obesity [21].

Our results showed a sex-related difference in the patient group in neurovegetative symptoms, which were more common among females. Still under the hypothesis of a role of inflammation, other studies have reported differences according to sex, notably in somatic symptoms in acute coronary heart disease [22]. In obese patients, inflammation was associated with depressive symptoms solely in the male participants [21] and in a model of inflammation induced by lipopolysaccharide, the women had more cytokines and more anxiety symptoms [41]. These findings suggest there are biological differences between women and men in the relationship between inflammation and psychiatric symptoms that could be explained by differences observed in inflammation [45, 46] and in depression and anxiety [47].

Our study also showed that the occurrence of psychiatric disorders was associated with high baseline symptom scores and with past psychotropic treatment. As mentioned above, the patients who were prescribed psychotropic drugs to alleviate their symptoms were nevertheless not suffering from more numerous psychiatric disorders nor had greater recourse to psychiatric consultations. A more detailed analysis shows that these patients had received more hypnotic drugs, in all likelihood for sleep disturbance, which is a neurovegetative symptom, and more antidepressants, which are effective against depressive, anxious and somatic symptoms [18]. Immunotherapy studies have shown that the development of depressive and anxious disorders are related to high depressive and anxious symptoms scores before the initiation of treatment [37, 38, 48]. These findings were confirmed in the present observational study of SRL patients. They also support the hypothesis of the "switch in vulnerable patients", in which depressive, neurovegetative and anxious symptoms related at the outset to inflammation subsequently lead to psychiatric disorders [12]. In chronic diseases, depression is a factor associated with poor quality of life, exacerbation of symptoms and increased hospital admissions and mortality rates [11]. It is therefore necessary to identify patients at risk and to tailor their management so as to limit the occurrence of psychiatric disorders. Antidepressants, which are effective against depressive, anxious and somatic symptoms [18], could achieve this aim and at the same time, probably owing to their anti-inflammatory action, exert a beneficial effect on dermatological disorders [49, 50]. Another study also showed that the CRP level could be taken into consideration in the choice of antidepressant [51]. These findings show that psychiatric disorders are not always associated with an increase in inflammation but, if that is the case, particularly in certain chronic inflammatory diseases, their management and treatment need to be adapted. The results of the present study are in line with those from the works cited above: high psychic anxiety symptoms scores were associated with past psychotropic treatment (mainly anxiolytics and hypnotics that are mostly prescribed by general practitioners in France) and the occurrence of psychiatric disorders during the follow-up period. However, only two of the nine patients who developed a psychiatric disorder had been receiving antidepressant drug.

Our post-hoc analysis hypothesized that the presence of psychiatric symptoms in SRL patients could have been due to the effects of inflammation but the initial cohort study had not

included a biological assay thus it was not possible to study such link via a biological analysis. However, our study was performed in the context of chronic inflammatory disease with increased cytokine production [1] and no other dermatological factor tested was associated with the different symptoms.

Many studies in the past used only overall scale scores and numerous others depression scales: publications on anxiety are rarer and more recent. In addition, many scales are designed to measure depressive and anxious symptoms but they do not all rate the same ones, and the symptoms are not always classified in the same manner. In our study, for example, we designated sleep problems, tiredness and changes in appetite as neurovegetative symptoms whereas Jokela et al. referred to them as "specific symptoms of depression" [39]. These discrepancies make it difficult to meaningfully compare studies. We used the MADRS for depressive and neurovegetative symptoms and the HAMA scale for psychic anxiety and somatic symptoms so as to cover all the symptoms as widely as possible using validated tools. Only the cognitive domain could not be studied, owing to the lack of data (there was just one cognitive item on each scale).

Because of the limited number of patients enrolled in the study it was not possible to perform multivariate analysis of the factors associated with the occurrence of psychiatric disorders during the follow-up. Nevertheless, the results were sufficient to evidence a greater presence of symptoms in the SRL patients with no axis 1 psychiatric disorders than in the healthy matched controls.

Besides the limited number of SRL patients enrolled in the study, we note differences at baseline between patients and controls concerning current personality disorders and past anxiolytic and hypnotic treatments. Moreover, the follow-up was not identical for all the patients since only 76% of them attended the last visit, which could have led to an underestimation of the development of axis I psychiatric disorders; however, it is important to emphasize that there were no differences between them regarding the MADRS and HAMA scores at baseline.

In conclusion, this study shows that SRL patients who were not suffering from current active axis I psychiatric disorders nevertheless experienced numerous neurovegetative, somatic and psychic anxiety symptoms. Those with marked symptoms of psychic anxiety were at risk of developing ulterior psychiatric disorders. This particular patient group should be closely monitored and their subclinical symptoms carefully managed before they progress towards a full psychiatric disorder.

## Supporting information

**S1 Fig. Patients' MADRS and HAMA scores during the follow-up.** Patients with high MADRS scores (extreme values) were not the same across visits.
(DOCX)

**S1 Table. Factors affecting symptom scores in participants.** Symptoms scores are presented as median [interquartile range]. [†] For age, spearman rho coefficients are presented. *Significantly different, p<0.05; ** p<0.01 and *** p<0.001.
(DOCX)

**S2 Table. Factors affecting symptom scores in patients.** Symptoms scores are presented as median [interquartile range]. [†] Spearman rho coefficients are presented. *Significantly different, p<0.05; ** p<0.01 and *** p<0.001.
(DOCX)

**S3 Table. Factors associated with occurrence of psychiatric disorders in skin-restricted lupus patients.** Results are presented as N (%), mean ± SD or median [inter quartile range].

Bold values indicate significant results with a p-value < 0.05. [†] 3 missing values. [††] Value for 21 patients. [†††] A total of 23 areas were considered: face, neck, collar, ears, scalp, anterior and posterior sides of the chest, abdomen, lumbar area, left and right arms, left and right forearms, palmar sides of left and right hands, dorsal sides of left and right hands, left and right thighs, left and right legs, dorsal sides of left and right feet.
(DOCX)

## Acknowledgments

The authors thank J. Watts for advice on the English version of the manuscript and gratefully acknowledge the contributions of the LuPsy Cohort investigators:

François Aubin (Franche Comté University, Besançon University Hospital, Department of Dermatology, Besançon, France), Christophe Bedane (Department of Dermatology, Hopital Dupuytren, Limoges, France), Sophie Bonnefond (Department of Psychiatric Emergencies, Esquirol Hospital, Limoges, France), Myriam Chastaing (Department of Liaison Psychiatry and Department of Dermatology, Brest University Hospital, Brest, France), Marianne Collange (Department of Adult Psychiatry and Medical Psychology, Clermont-Ferrand University Hospital, Clermont-Ferrand, France), Patrick Combemale, Denys Courbier (Desgenettes Army Training Hospital, Lyon, France), Carole Durand (Department of Adult Psychiatry and Medical Psychology, Clermont-Ferrand University Hospital, Clermont-Ferrand, France), Jean Paul Grand (Le Valmont Hospital, Emergencies and Liaison Psychiatry, Valence General Hospital, Valence, France), Emmanuel Haffen (Department of Clinical Psychiatry, INSERM, University hospital of Besançon, Besançon, France), Bruno Labeille (Department of Dermatology, Nord University Hospital, Saint-Etienne, France), Catherine Massoubre (Department of Psychiatry, St-Etienne University Hospital, University Jean Monnet, St-Etienne, France), Laurent Misery (University Hospital of Brest, Department of Dermatology and University of Western Brittany, Laboratory of Neurosciences of Brest, Brest, France), Jean-Luc Perrot (Department of Dermatology, Nord University Hospital, Saint-Etienne, France), Anne Laure Pontonnier (Department of Adult Psychiatry and Medical Psychology, Clermont-Ferrand University Hospital, Clermont-Ferrand, France), Robert Schwan (University Hospital of Psychiatry and Psychotherapy, Laxou, France), François Skowron (Department of Dermatology, Valence Hospital, Valence, France), Agnès Sparsa (Department of Internal Medicine, Clinic Mutualiste Catalane, Perpignan, France), Gaëlle Theilhol (Department of Adult Psychiatry and Medical Psychology, Clermont-Ferrand University Hospital, Clermont-Ferrand, France), Julie Waton (Department of Dermatology, Nancy University Hospital, Nancy, France).

## Author Contributions

**Conceptualization:** Fabien Rondepierre, Michel D'Incan, Sophie Lauron, Isabelle Jalenques.

**Data curation:** Urbain Tauveron-Jalenques, Solène Valette.

**Formal analysis:** Fabien Rondepierre, Aurélien Mulliez.

**Funding acquisition:** Michel D'Incan, Isabelle Jalenques.

**Investigation:** Sophie Lauron.

**Methodology:** Fabien Rondepierre, Aurélien Mulliez, Sophie Lauron.

**Project administration:** Fabien Rondepierre.

**Resources:** Michel D'Incan.

**Software:** Fabien Rondepierre.

**Supervision:** Sophie Lauron, Isabelle Jalenques.

**Validation:** Isabelle Jalenques.

**Writing – original draft:** Fabien Rondepierre, Isabelle Jalenques.

**Writing – review & editing:** Aurélien Mulliez, Michel D'Incan, Sophie Lauron.

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
