## [Decision Letter · Decision Letter 0]

31 May 2022

PONE-D-21-34055Psychiatric symptomatology in skin-restricted lupus patients without psychiatric disorders: a post-hoc analysisPLOS ONE

Dear Dr. Jalenques,

Thank you for submitting your manuscript to PLOS ONE. After careful consideration, we feel that it has merit but does not fully meet PLOS ONE’s publication criteria as it currently stands. Therefore, we invite you to submit a revised version of the manuscript that addresses the points raised during the review process. Please note that we have only been able to secure a single reviewer to assess your manuscript. We are issuing a decision on your manuscript at this point to prevent further delays in the evaluation of your manuscript. Please be aware that the editor who handles your revised manuscript might find it necessary to invite additional reviewers to assess this work once the revised manuscript is submitted. However, we will aim to proceed on the basis of this single review if possible.  Your manuscript has been assessed by an expert reviewer, whose comments are appended below. The reviewer has highlighted concerns about some aspects of the methodology and made some suggestions for improving the framing of research question and discussion. Please ensure you respond to each point carefully in your response to reviewers document, and modify your manuscript accordingly.

We look forward to receiving your revised manuscript.

Kind regards,

Joseph Donlan

Editorial Office

PLOS ONE

Journal Requirements:

Reviewers' comments:

Reviewer's Responses to Questions

**Comments to the Author**

1. Is the manuscript technically sound, and do the data support the conclusions?

Reviewer #1: Yes

2. Has the statistical analysis been performed appropriately and rigorously? 

Reviewer #1: Yes

3. Have the authors made all data underlying the findings in their manuscript fully available?

Reviewer #1: Yes

4. Is the manuscript presented in an intelligible fashion and written in standard English?

Reviewer #1: Yes

5. Review Comments to the Author

Reviewer #1: Thank you for the opportunity to review this manuscript. Please see below for my comments, which I hope prove useful.

Introduction

Suggest making it clear that the aim of the study is to investigate the relationship between depressive and anxiety symptoms in patients with SRL without PREVALENT psychiatric disorders. I understand that there might be participants included in this analysis who have a history of depression or an anxiety disorder

Methods

Please provide a brief statement about how controls were recruited

Suggest briefly explaining the rationale for using the MADRS and HAMA measures. There measures are a major component of the analyses. Please also describe the rationale for creating the subscales (depressive, neurovegetative, psychic anxiety and somatic symptom scores) from these measures. As a psychiatrist, the terms are a bit confusing, as “depressive” symptoms typically include neurovegetative, anxiety and somatic symptoms. Perhaps a better term would be “depressive cognitions and affect” or something to that effect

How were previous psychiatric disorders diagnosed? Also via the MINI or by self-report?

Please provide the rationale for excluding current Axis 1 psychiatric disorders, but not personality disorders

It is not clear to me exactly how the variables included in the factor analysis (comparing patients with and without incident psychiatric disorders) were selected. This section would benefit from more explanation for transparency

Results

The text explaining Figure 2 is very difficult to interpret. Suggest explaining the figure in more detail. I know it represents a factor analysis, but find it hard to understand

Boxplots need labels on the X and Y axes and Fig 2 needs a legend

Discussion

As alluded to above, it is very difficult to determine whether participants may have had past psychiatric diagnoses retrospectively. The method used in this study for diagnosing past psychiatric disorders would help to explain the discrepancy between past psychotropic medication use and past psychiatric disorder in the findings.

I wasn’t sure what this paragraph was referring to (page 18):

“Our study hypothesized that the presence of psychiatric symptoms in SRL patients could have been due to the effects of inflammation but post-hoc analysis without inflammation assays failed to establish any such link. However, our study was performed in the context of chronic inflammatory disease with increased cytokine production [1] and no other dermatological factor tested was associated with the different symptoms.”

Where was this analysis done?

6. PLOS authors have the option to publish the peer review history of their article (what does this mean?). If published, this will include your full peer review and any attached files.

Reviewer #1: No

---

## [Author Response · Author response to Decision Letter 0]

23 Jun 2022

Response to the academic editor

Style requirement has been checked and changed when required.

All the data are now available at Mendeley:

Rondepierre, Fabien; Tauveron-Jalenques, Urbain; Valette, Solène; Mulliez, Aurélien; D'Incan, Michel; Lauron, Sophie; Jalenques, Isabelle (2022), “Psychiatric symptomatology in skin-restricted lupus patients without psychiatric disorders: a post-hoc analysis”, Mendeley Data, V1, doi: 10.17632/bdxf94z5xw.1

While revising your submission, please upload your figure files to the Preflight Analysis and Conversion Engine (PACE) digital diagnostic tool, https://pacev2.apexcovantage.com/. PACE helps ensure that figures meet PLOS requirements.

All our figures have been checked with PACE digital diagnostic tool and they meet PLOS requirements.

 

Response to reviewers

Reviewer #1: Thank you for the opportunity to review this manuscript. Please see below for my comments, which I hope prove useful.

Introduction

1.Suggest making it clear that the aim of the study is to investigate the relationship between depressive and anxiety symptoms in patients with SRL without PREVALENT psychiatric disorders. I understand that there might be participants included in this analysis who have a history of depression or an anxiety disorder

We don't usually use "prevalent psychiatric disorders", do you mean “current psychiatric disorders”?

If this is the case, we agree that it is important to specify it. This has been added at the end of the introduction: “In the present exploratory study, we used longitudinal data to investigate the presence of depressive, psychic anxiety, neurovegetative and somatic symptoms in SRL patients without current psychiatric disorders.”

Methods

2.Please provide a brief statement about how controls were recruited

We agree that the reference to previous articles is not sufficient (references 3 and 21) and a description of the recruitment of control subjects has been added: “Data from the studies of the LuPsy cohort were used for this analysis. This cohort comprised 80 consecutive outpatients with chronic SRL (DLE and LT) or SCLE and 160 healthy control subjects without a history of lupus recruited among volunteers from a clinical research centre and workers in public hospitals, the national railway company and administrative personnel of the state education system.”

3.Suggest briefly explaining the rationale for using the MADRS and HAMA measures. There measures are a major component of the analyses. Please also describe the rationale for creating the subscales (depressive, neurovegetative, psychic anxiety and somatic symptom scores) from these measures. As a psychiatrist, the terms are a bit confusing, as “depressive” symptoms typically include neurovegetative, anxiety and somatic symptoms. Perhaps a better term would be “depressive cognitions and affect” or something to that effect

The MADRS and HAMA scales are those used in the LuPsy cohort whose data we used again. These scales have the advantage of scanning the various possible symptoms.

As you noticed and as indicated at the end of our discussion in the limit part, symptoms are not always classified in the same manner. These different subscales were previously used in research by numerous authors and we used the same . In addition, this classification is important since we wanted to know which symptom was represented in lupus patients without current psychiatric disorders and which could be associated with the development of psychiatric disorder. Thus we could study if a parallel was possible with the model of cytokine-induced depression (since lupus is an inflammatory disease) with early-onset neurovegetative and somatic symptoms of depression and late-onset psychological symptoms of depression that include mild cognitive alterations and symptoms of depressed mood, sometimes accompanied by anxiety.

4.How were previous psychiatric disorders diagnosed? Also via the MINI or by self-report?

Indeed past psychiatric disorders were also diagnosed via the MINI. This precision has been added in the method part.

5.Please provide the rationale for excluding current Axis 1 psychiatric disorders, but not personality disorders

We agree that there are links between depressive and anxiety disorders and personality disorders. We wanted to study the symptoms present in skin-restricted lupus patients without psychiatric disorders; and also to study whether those patients who present with symptoms have more evolution or not towards a disorder. 

We did not exclude personality disorders because a personality disorder is an enduring pattern of inner experience and behavior that deviates markedly from the individual’s culture, is pervasive and inflexible, has an onset in adolescence or early adulthood and is stable over time (DSM).

Our results, presented in table S2, showed that only the depressive symptom score was related to the presence of a current personality disorder in patients. Additionaly the occurrence of a psychiatric disorder during the follow-up period was mostly associated with higher psychic anxiety symptom scores at baseline (Fig 3 and S3 Table); but it was not associated with personality disorders. 

6.It is not clear to me exactly how the variables included in the factor analysis (comparing patients with and without incident psychiatric disorders) were selected. This section would benefit from more explanation for transparency

As indicated, the variables considered in this analysis were determined based on the univariate results and clinical relevance among sociodemographic, dermatological and psychiatric data. Thus the HAMA scores and the presence of past psychotropic medication showed difference between patients who developed disorder and those who did not. All other included variables were those that could impact the development of psychiatric disorder on the basis of author’s experience, clinical relevance and bibliographic data such the age and the gender of the patients, the dermatological data (number and size of the lesions, presence of a treatment, type of lupus, presence of pruritus, smoker status) and psychiatric data (history, medication, consultation). As the objective of our work was to investigate the presence and impact of specific symptoms, all HAMA and MADRS symptom scores have been added to the analysis.

Clarifications have been made to this part.

Results

7.The text explaining Figure 2 is very difficult to interpret. Suggest explaining the figure in more detail. I know it represents a factor analysis, but find it hard to understand

We agree that this figure is difficult to interpret and a more detailed explanation has been added: “The Fig 2A showed the correlation circle of the variables used in the factor analysis on mixed data. The first axis, which represented 24.7% of the variance of the data considered in the analysis, was more linked to the sex, HAMA and MADRS scores, whereas the second axis, with 15.3% of the variance, was more associated with the presence of past psychiatric disorder or lupus or psychiatric medications. The Fig 2B revealed that the patients who developed a psychiatric disorder during the follow-up had a distinctive baseline profile than those who did not develop such disorders. These two profiles were mainly marked by different HAMA scores, with higher scores for the patients who developed a psychiatric disorder during the follow-up.”

8.Boxplots need labels on the X and Y axes and Fig 2 needs a legend

Labels on the X and Y axes have been added on the figures 1 and 3. The figure 2 has been edited for clarity and its legend modified.

Discussion

9.As alluded to above, it is very difficult to determine whether participants may have had past psychiatric diagnoses retrospectively. The method used in this study for diagnosing past psychiatric disorders would help to explain the discrepancy between past psychotropic medication use and past psychiatric disorder in the findings.

The search for past psychiatric disorders was carried out through a psychiatric interview using the MINI, a method which is considered the gold standard. However, we cannot exclude defects in the recall of the disorders, or the psychotropic treatments or the psychiatrist consultations. But the discrepancy between past psychotropic medication use and past psychiatric disorder could above all be explained by the fact that in France psychotropic medications are often prescribed by general practitioners when people complain of psychiatric symptoms even if the necessary criteria are not completely fulfilled to make the diagnosis of psychiatric disorder.

The possibility of recall bias has been added to the discussion on page 16.

10.I wasn’t sure what this paragraph was referring to (page 18):

“Our study hypothesized that the presence of psychiatric symptoms in SRL patients could have been due to the effects of inflammation but post-hoc analysis without inflammation assays failed to establish any such link. However, our study was performed in the context of chronic inflammatory disease with increased cytokine production [1] and no other dermatological factor tested was associated with the different symptoms.”

Where was this analysis done?

We thank you for this remark which leads us to rephrase for more clarity. Our post-hoc analysis hypothesized that the presence of psychiatric symptoms in SRL patients could have been due to the effects of inflammation but the initial cohort study had not included a biological assay thus it was not possible to study such link via a biological analysis.

This sentence has been rewritten for clarity.

---

## [Decision Letter · Decision Letter 1]

14 Sep 2022

PONE-D-21-34055R1Psychiatric symptomatology in skin-restricted lupus patients without psychiatric disorders: a post-hoc analysisPLOS ONE

Dear Dr. Jalenques,

Thank you for submitting your manuscript to PLOS ONE. After careful consideration, we feel that it has merit but does not fully meet PLOS ONE’s publication criteria as it currently stands. Therefore, we invite you to submit a revised version of the manuscript that addresses the points raised during the review process.

Unfortunately, the reviewer who commented on your original submission was not available to review your revised manuscript. However, we have obtained feedback from three additional external reviewers. They have identified a number of outstanding concerns that will need to be carefully addressed in a further revision. Please respond to all of the points the reviewers have raised when preparing your revision.

We look forward to receiving your revised manuscript.

Kind regards,

Jamie Males

Editorial Office

PLOS ONE

Reviewers' comments:

Reviewer's Responses to Questions

**Comments to the Author**

1. If the authors have adequately addressed your comments raised in a previous round of review and you feel that this manuscript is now acceptable for publication, you may indicate that here to bypass the “Comments to the Author” section, enter your conflict of interest statement in the “Confidential to Editor” section, and submit your "Accept" recommendation.

Reviewer #2: (No Response)

Reviewer #3: All comments have been addressed

Reviewer #4: All comments have been addressed

2. Is the manuscript technically sound, and do the data support the conclusions?

Reviewer #2: (No Response)

Reviewer #3: Partly

Reviewer #4: Yes

3. Has the statistical analysis been performed appropriately and rigorously? 

Reviewer #2: (No Response)

Reviewer #3: No

Reviewer #4: Yes

4. Have the authors made all data underlying the findings in their manuscript fully available?

Reviewer #2: (No Response)

Reviewer #3: No

Reviewer #4: Yes

5. Is the manuscript presented in an intelligible fashion and written in standard English?

Reviewer #2: (No Response)

Reviewer #3: Yes

Reviewer #4: Yes

6. Review Comments to the Author

Reviewer #2: Thank you for the opportunity to review this manuscript.

This paper aims to investigate the subclinical psychiatric symptoms in skin-restricted lupus patients without psychiatric disorders and to study the occurrence of psychiatric disorders during the study follow-up. The study shows that skin-restricted lupus patients experience depression and anxiety symptoms and that those with marked symptoms are at risk of developing psychiatric disorders. The major limitation of the study is that the group of patients had twice as many current personality disorders and twice as much past use of psychotropic medications compared to controls. Those variables are shown in the study to be associated with their subclinical psychiatric symptoms in addition to their cutaneous lupus. The particular contribution of their skin disease to their psychiatric symptoms is unclear.

Please see below for my detailed comments:

I. Title:

• It seems that the term “patients without psychiatric disorders” does not define properly this group of patients who have a high prevalence of personality disorders that is much higher than in the general population. I suggest refining this term.

II. Abstract:

• Although the authors’ aim is to investigate anxiety and depressive symptoms in skin-restricted lupus patients without psychiatric disorders the patients had a higher prevalence of psychiatric morbidity including current personality disorder and a history of psychotropic medications than controls, which cannot be overlooked. The abstract should emphasize the differences between groups either in the results or the limitations section.

• The results should include the multivariate analysis as well that shows that current personality disorder and a history of psychotropic medications were associated with higher MADRS and HAMA scores as well.

• The second aim of the study regarding the occurrence of psychiatric disorders during the study follow-up (as mentioned in the results section of the abstract) should be included in the abstract.

III. Introduction:

• The authors note the connection between depression and anxiety and chronic inflammatory diseases and list diseases “such as psoriasis, alopecia areata, atopic dermatitis, systemic lupus erythematosus, inflammatory bowel disease, multiple sclerosis and rheumatoid arthritis” but subsequently bring examples from depression and anxiety observed in obesity and coronary heart disease which are not classical inflammatory diseases. Instead, citing the literature about depression and anxiety in classic chronic inflammatory diseases is reasonable.

• There is no explanation and no reference to the statement that: “It would be useful, therefore, to identify specific symptoms associated with chronic inflammatory disease”. Relevant literature about the importance of identifying subclinical depression and anxiety including the usefulness of interventions in preventing progression may be of interest.

IV. Discussion:

• The authors indicate tumor necrosis factor (TNF) antagonist treatment, as a potential treatment for depression in patients with elevated CRP and mention its role in treating dermatologic patients. It is not clear what the authors tried to say. Do they imply the relevance to cutaneous lupus? Although TNF antagonists are indeed used in some chronic inflammatory dermatologic diseases like psoriasis, they have no role in treating cutaneous lupus. This issue should be cleared.

• The discussion should include major limitations including the small number of patients and The basic differences between patients and controls which might confound the results albeit the multivariate analysis (e.g. history of psychotropic medications, current personality disorder).

V. Conclusion:

• By reading the conclusion the impression made is that the patients have no psychiatric morbidity neither currently nor in the past which is not the case. The sentence should be caveated to convey the presence of more personality disorders and more past psychotropic treatment.

Reviewer #3: Dear Authors and Editors,

1. 5 patients took Antidepressant and 4 took Anxiolytics but they had no Axis I diagnosis. Authors should clearly report what antidepressants/anxiolytics they used for which reasons. Otherwise, it may be conflicting with their inclusion criteria.

2. 12 controls were on Current psychiatrist consultation. 10 controls took Antidepressants and 14 took Anxiolytic. The control selection was inappropriate because those with skin-limited LE having current Axis 1 diagnosis were excluded from the case sample. Those with current Axis 1 diagnosis should be excluded in the control sample.

3. I am very surprised on the prevalence of personality disorder. 32% was so high. Authors should clearly report cluster A, B, C PD prevalence, separately. In addition, I am also very surprised at the great gap between PD diagnosis and no current Axis 1 diagnosis in the case sample.

4. Authors should clearly report what Medical comorbidities are.

5. Table 2 showed selected associations between Associated factors with symptom scores in all participants. Why did authors only report 3 factors? How about other factors? such as Current psychotropic treatment...?

6. This study seemed to be a follow-up study. But, the related information was very limited in the method section. How long were those participants followed? When was the second assessment? Were controls followed?

7. In the results, "The main factors identified were.....had higher depressive and psychic anxiety symptom scores than patients without". This analysis was done based on all participants. I supposed some were not patients.

8. "....the other 33 underwent 4.2 assessments on average, with an average follow-up of 21.9 ± 5.9 months."...Did it mean each patient were followed with different duration and at different time? Is it a standard cohort study? Why was the follow-up timepoint so different?

9. "The Fig 2B revealed that the patients who developed a psychiatric disorder during the follow-up had a distinctive baseline profile than those who did not develop such disorders."...Psychiatric disorder has many diagnoses...So, what was the definite diagnosis in such patients who developed a psychiatric disorder? Was it reasonable to compare the baseline difference between patients with or without a psychiatric disorder because a psychiatric disorder was so heterogenous?

Reviewer #4: The Authors have addressed all points raised by the referee(s). The manuscript (including the tables/figures) have been revised accordingly.

7. PLOS authors have the option to publish the peer review history of their article (what does this mean?). If published, this will include your full peer review and any attached files.

Reviewer #2: No

Reviewer #3: No

Reviewer #4: No

---

## [Author Response · Author response to Decision Letter 1]

28 Oct 2022

PONE-D-21-34055R1

Psychiatric symptomatology in skin-restricted lupus patients without psychiatric disorders: a post-hoc analysis

PLOS ONE

Response to reviewers

6. Review Comments to the Author

Reviewer #2: Thank you for the opportunity to review this manuscript.

This paper aims to investigate the subclinical psychiatric symptoms in skin-restricted lupus patients without psychiatric disorders and to study the occurrence of psychiatric disorders during the study follow-up. The study shows that skin-restricted lupus patients experience depression and anxiety symptoms and that those with marked symptoms are at risk of developing psychiatric disorders. The major limitation of the study is that the group of patients had twice as many current personality disorders and twice as much past use of psychotropic medications compared to controls. Those variables are shown in the study to be associated with their subclinical psychiatric symptoms in addition to their cutaneous lupus. The particular contribution of their skin disease to their psychiatric symptoms is unclear.

Please see below for my detailed comments:

I. Title:

• It seems that the term “patients without psychiatric disorders” does not define properly this group of patients who have a high prevalence of personality disorders that is much higher than in the general population. I suggest refining this term.

In order to be more accurate, we have changed “patients without psychiatric disorders” to “patients without axis I psychiatric disorder” throughout the text.

II. Abstract:

• Although the authors’ aim is to investigate anxiety and depressive symptoms in skin-restricted lupus patients without psychiatric disorders the patients had a higher prevalence of psychiatric morbidity including current personality disorder and a history of psychotropic medications than controls, which cannot be overlooked. The abstract should emphasize the differences between groups either in the results or the limitations section.

The differences between lupus patients and controls have been added in the result section of the abstract: “None of the participants had any current axis I psychiatric disorders but the patients had personality disorders more frequently and had received more past psychotropic treatments than the controls.”

• The results should include the multivariate analysis as well that shows that current personality disorder and a history of psychotropic medications were associated with higher MADRS and HAMA scores as well.

The results of the multivariate analysis have been included in the abstract: “No dermatological factor tested was associated with these scores. , whereas being a lupus patient was associated with higher neurovegetative and somatic symptoms scores, having a current personality disorder with higher depressive and neurovegetative scores and receiving more past psychotropic treatments with psychic anxiety and somatic symptoms scores.”

• The second aim of the study regarding the occurrence of psychiatric disorders during the study follow-up (as mentioned in the results section of the abstract) should be included in the abstract.

This second aim has been added to the abstract: “It was decided, therefore, to investigate the presence of specific symptoms in skin-restricted lupus patients without axis I psychiatric disorders and their impact on the occurrence of axis I psychiatric disorders during the study follow-up.”

III. Introduction:

• The authors note the connection between depression and anxiety and chronic inflammatory diseases and list diseases “such as psoriasis, alopecia areata, atopic dermatitis, systemic lupus erythematosus, inflammatory bowel disease, multiple sclerosis and rheumatoid arthritis” but subsequently bring examples from depression and anxiety observed in obesity and coronary heart disease which are not classical inflammatory diseases. Instead, citing the literature about depression and anxiety in classic chronic inflammatory diseases is reasonable.

At the end of the introduction, we have added references about depression and anxiety in classic chronic inflammatory diseases. “Since then, many studies have looked for links between inflammation and depressive symptoms, including in chronic inflammatory diseases [4,6,8,9,19]” .

Regarding obesity, diabetes and coronary heart disease which are not classical inflammatory diseases, but are associated with inflammation we have edited the sentence and started a new paragraph to avoid confusion and improve clarity: Additionnally, inflammation and depressive symptoms were observed in association in obese men and in diabetic patients [20,21]. In acute coronary heart disease patients with no history of depression, inflammation was associated with somatic symptoms [22]. In the same study, a longitudinal association was also found between inflammation and anxious symptoms [22].”

• There is no explanation and no reference to the statement that: “It would be useful, therefore, to identify specific symptoms associated with chronic inflammatory disease”. Relevant literature about the importance of identifying subclinical depression and anxiety including the usefulness of interventions in preventing progression may be of interest.

We cited references about the importance of identifying symptoms in the previous sentence: “These subclinical disorders could be risk factors for progression towards genuine depression and anxiety disorders [13,15] ”

We have added a sentence and reference on the effectiveness of symptom management in preventing progression: “Additionally, interventions in preventing progression may be of interest [16]” .

IV. Discussion:

• The authors indicate tumor necrosis factor (TNF) antagonist treatment, as a potential treatment for depression in patients with elevated CRP and mention its role in treating dermatologic patients. It is not clear what the authors tried to say. Do they imply the relevance to cutaneous lupus? Although TNF antagonists are indeed used in some chronic inflammatory dermatologic diseases like psoriasis, they have no role in treating cutaneous lupus. This issue should be cleared.

For clarity, we have removed this passage a little further from the discussion.

• The discussion should include major limitations including the small number of patients and The basic differences between patients and controls which might confound the results albeit the multivariate analysis (e.g. history of psychotropic medications, current personality disorder).

We added limitations taking up these elements at the end of the discussion: “Besides the limited number of SRL patients enrolled in the study, we note differences at baseline between patients and controls concerning current personality disorders and past anxiolytic and hypnotic treatments. Moreover, the follow-up was not identical for all the patients since only 76% of them attended the last visit, which could have led to an underestimation of the development of axis I psychiatric disorders; however, it is important to emphasize that there were no differences between them regarding the MADRS and HAMA scores at baseline.”

V. Conclusion:

• By reading the conclusion the impression made is that the patients have no psychiatric morbidity neither currently nor in the past which is not the case. The sentence should be caveated to convey the presence of more personality disorders and more past psychotropic treatment.

We changed the sentence accordingly: “In conclusion, this study shows that SRL patients who were not suffering from current axis I psychiatric disorders nevertheless experienced numerous neurovegetative, somatic and psychic anxiety symptoms. Those with marked symptoms of psychic anxiety were at risk of developing ulterior psychiatric disorders.”

Reviewer #3: Dear Authors and Editors,

1. 5 patients took Antidepressant and 4 took Anxiolytics but they had no Axis I diagnosis. Authors should clearly report what antidepressants/anxiolytics they used for which reasons. Otherwise, it may be conflicting with their inclusion criteria.

One patient with “anxiolytic” was corrected in “hypnotic”. For the others, it is the same patients who have antidepressants and anxiolytics with long-term treatments (between 2.5 and 30 years of duration). Indications for treatment were depression (n=3) and anxiety disorders (one phobia and one panic disorder). Even if some treatments do not seem to comply with the recommendations, a psychiatrist checked that the patients did not have current axis I psychiatric disorders at the time of inclusion. Lastly, it should be noted that even if current psychotropic treatment is associated with high scores of psychic and somatic anxiety, taking treatment is not associated with the occurrence of psychiatric disorders during follow-up.

2. 12 controls were on Current psychiatrist consultation. 10 controls took Antidepressants and 14 took Anxiolytic. The control selection was inappropriate because those with skin-limited LE having current Axis 1 diagnosis were excluded from the case sample. Those with current Axis 1 diagnosis should be excluded in the control sample.

We thank you for noticing this error and we apologize for it. Regarding the controls, they had no current psychotropic treatment; these are the past psychotropic treatments which had been mistakenly reported in "current psychotropic treatment" (see table 1) 

3. I am very surprised on the prevalence of personality disorder. 32% was so high. Authors should clearly report cluster A, B, C PD prevalence, separately. In addition, I am also very surprised at the great gap between PD diagnosis and no current Axis 1 diagnosis in the case sample.

We have reported cluster A, B, C PD prevalence, separately in table 1. There is no significant difference between patients and controls on the different cluster PD prevalence, even if patients have a little more cluster A and C PD prevalence. With regard to PD, the prevalence that we observed in our study is close to that found in other studies using the PDQ4 with the clinical significance scale (27% Bouvard; 38% Calvo) . A PD is relatively stable over time (DSM 5). Although patients may not have current axis I psychiatric disorders, 50% of patients have had past psychiatric disorders. If we look at all the participants, 59% of those with a PD have already had an axis I psychiatric disorder.

4. Authors should clearly report what Medical comorbidities are.

The medical comorbidities were as follows:

 Patients, n (%) Controls, n (%)

Infection 2 (5) 1 (1)

Tumor 3 (8) 7 (9)

Endocrine, nutritional or metabolic diseases 3 (8) 6 (8)

Circulatory system 8 (21) 11 (14)

Digestive system 4 (11) 8 (11)

Musculoskeletal system and connective tissue 5 (13) 6 (8)

Genitourinary system 2 (5) 2 (3)

Nervous system 2 (5) 4 (5)

Respiratory system 3 (8) 1 (1)

Diseases of eyes 2 (5) 1 (1)

Diseases of the ear 1 (3) 0

In order to avoid overburdening the table, the main comorbidities have been added at the bottom of the note in Table 1.

5. Table 2 showed selected associations between Associated factors with symptom scores in all participants. Why did authors only report 3 factors? How about other factors? such as Current psychotropic treatment...?

Given the number of participants, the number of variables that can be included in the multivariate model is limited and we must ensure that there is no collinearity between the variables.

We only reported the factors associated with the scores in the univariate analysis (see Table S1 in the supplementary data). This was already specified in "Statistical method" (“Multivariate analyses of factors associated with the symptom scores were performed by multiple generalized linear regression, taking the group (patients vs controls) as main covariate and selecting other covariates, based on clinically relevant factors or statistically significant factors shown in univariate analysis”), then in the “Results” (“To understand why patients had higher scores than controls and to identify the factors responsible for the differences, we performed multiple generalized linear regression using the study population (patients vs controls) and the previously identified factors (current personality disorders and past psychotropic treatment) as covariates. The results are presented in table 2.”). 

New results were obtained following the correction concerning “psychotropic treatments”: current psychotropic treatment is associated with higher anxiety and somatic scores (see S1 Table). We therefore added current psychotropic treatment to the multivariate model (after checking that there was no collinearity between the variables). Scores aren’t associated with current psychotropic treatment. Only one change is observed: a stronger association between being a patient and the somatic anxiety score (p=0.09).

6. This study seemed to be a follow-up study. But, the related information was very limited in the method section. How long were those participants followed? When was the second assessment? Were controls followed?

We agree that relying on previous papers for the description of the LuPsy cohort is not sufficient; we have therefore added a brief description in the “Participants” part of the “Method” section: “The patients were followed for 2 years with a visit every 6 months (5 visits in total) including both a dermatological and a psychiatric evaluation. Controls had no follow-up.”

7. In the results, "The main factors identified were.....had higher depressive and psychic anxiety symptom scores than patients without". This analysis was done based on all participants. I supposed some were not patients.

We thank you for noticing this error and we apologize for it. The term “patients” has been replaced by “participants” in the text.

8. "....the other 33 underwent 4.2 assessments on average, with an average follow-up of 21.9 ± 5.9 months."...Did it mean each patient were followed with different duration and at different time? Is it a standard cohort study? Why was the follow-up timepoint so different?

A dermatological and psychiatric evaluation was carried out every 6 +/-1 months. It is therefore possible that all the patients didn’t have exactly the same follow-up. Moreover, as indicated in the results, only 76% of patients attended the last visit.

Among patients who had a psychiatric disorder, follow-up was 20.4 +/- 6.9 months; among patients without a psychiatric disorder, follow-up was 22.4+/-5.6 months; there is no difference considering the standard deviation. 

This could have led to an underestimation of the development of axis I psychiatric disorders. 

Those lost to follow-up could be patients who have had a psychiatric disorder. However, it is important to emphasize that there were no differences regarding the MADRS and HAMA scores at baseline between the patients who attended the last visit and those who didn’t.

These points have been discussed in the limits.

“Besides the limited number of SRL patients enrolled in the study, we note differences at baseline between patients and controls concerning current personality disorders and past anxiolytic and hypnotic treatments. Moreover, the follow-up was not identical for all the patients since only 76% of them attended the last visit, which could have led to an underestimation of the development of axis I psychiatric disorders; however, it is important to emphasize that there were no differences between them regarding the MADRS and HAMA scores at baseline.”

9. "The Fig 2B revealed that the patients who developed a psychiatric disorder during the follow-up had a distinctive baseline profile than those who did not develop such disorders."...Psychiatric disorder has many diagnoses...So, what was the definite diagnosis in such patients who developed a psychiatric disorder? Was it reasonable to compare the baseline difference between patients with or without a psychiatric disorder because a psychiatric disorder was so heterogenous?

Psychiatric disorders that occurred during the follow-up are three depressive disorders (including one with anxiety disorder), two dysthymia, one suicide risk, two panic disorders and one social phobia.

Almost all these disorders are depressive or anxious disorders whose symptoms can be identified and measured on anxiety and depression scales such as MADRS and HAMA.

The description of the disorders has been added in the "Results" section.

Reviewer #4: The Authors have addressed all points raised by the referee(s). The manuscript (including the tables/figures) have been revised accordingly.

---

## [Decision Letter · Decision Letter 2]

11 Jan 2023

PONE-D-21-34055R2Psychiatric symptomatology in skin-restricted lupus patients without axis I psychiatric disorders: a post-hoc analysisPLOS ONE

Dear Dr. Jalenques,

Thank you for submitting your manuscript to PLOS ONE. After careful consideration, we feel that it has merit but does not fully meet PLOS ONE’s publication criteria as it currently stands. Therefore, we invite you to submit a revised version of the manuscript that addresses the points raised during the review process.

Please revise this paper and provide responses on the queries from reviewer 3.

We look forward to receiving your revised manuscript.

Kind regards,

Nasrin Akter, MPH

Guest Editor

PLOS ONE

Reviewers' comments:

Reviewer's Responses to Questions

**Comments to the Author**

1. If the authors have adequately addressed your comments raised in a previous round of review and you feel that this manuscript is now acceptable for publication, you may indicate that here to bypass the “Comments to the Author” section, enter your conflict of interest statement in the “Confidential to Editor” section, and submit your "Accept" recommendation.

Reviewer #2: All comments have been addressed

Reviewer #3: All comments have been addressed

Reviewer #4: All comments have been addressed

Reviewer #5: All comments have been addressed

2. Is the manuscript technically sound, and do the data support the conclusions?

Reviewer #2: (No Response)

Reviewer #3: No

Reviewer #4: Yes

Reviewer #5: Yes

3. Has the statistical analysis been performed appropriately and rigorously? 

Reviewer #2: (No Response)

Reviewer #3: No

Reviewer #4: Yes

Reviewer #5: Yes

4. Have the authors made all data underlying the findings in their manuscript fully available?

Reviewer #2: (No Response)

Reviewer #3: No

Reviewer #4: Yes

Reviewer #5: Yes

5. Is the manuscript presented in an intelligible fashion and written in standard English?

Reviewer #2: (No Response)

Reviewer #3: Yes

Reviewer #4: Yes

Reviewer #5: Yes

6. Review Comments to the Author

Reviewer #2: (No Response)

Reviewer #3: Dear Authors and Editors,

1. For me, it is not very reasonable for the reply that "Indications for treatment were depression (n=3) and

anxiety disorders (one phobia and one panic disorder)" and "a psychiatrist checked that the patients did

not have current axis I psychiatric disorders at the time of inclusion". I suppose authors may indicate no active psychiatric episode, or remitted state of a psychiatric disorder. If no diagnosis, why did those patients take medications?

2. I am confused for the definition of Axis I and II diagnoses in current study. Is Axis I diagnosis based on a psychiatrist's interview? Is Axis II diagnosis based on the 99-item self-report Personality Diagnostic

Questionnaire? Axis I and Axis II are clinical diagnoses. I am not sure whether authors used different modality to define those diagnoses. In addition, based on DSM 5 (as authors reported), the concept of axis diagnosis is not applied.

3. "The psychiatrist also recorded scores from the Montgomery-Asberg Depression Rating Scale (MADRS) [30] and the Hamilton Anxiety Rating scale (HAMA) [31]"..Authors should report the symptom scores at baseline and during the follow-up between cases and controls.

4. based on figure 1, we can see several controls rated >= 10 on MADRS and even more cases rated >= 10 on MADRS. The remission criteria of MADRS is commonly defined at 10. Authors said a psychiatrist diagnosed no axis 1 diagnosis in cases and controls, which were conflicting with the rating scores.

Reviewer #4: I have read the responses to the reviewers' points as well as the revised manuscript which has been improved.

Reviewer #5: (No Response)

7. PLOS authors have the option to publish the peer review history of their article (what does this mean?). If published, this will include your full peer review and any attached files.

Reviewer #2: No

Reviewer #3: No

Reviewer #4: No

Reviewer #5: No

---

## [Author Response · Author response to Decision Letter 2]

2 Feb 2023

Reviewer #3: Dear Authors and Editors,

1. For me, it is not very reasonable for the reply that "Indications for treatment were depression (n=3) and anxiety disorders (one phobia and one panic disorder)" and "a psychiatrist checked that the patients did not have current axis I psychiatric disorders at the time of inclusion". I suppose authors may indicate no active psychiatric episode, or remitted state of a psychiatric disorder. If no diagnosis, why did those patients take medications?

We understand your comment. Indeed all these patients have a history of psychiatric disorders but no active psychiatric episode. Moreover, these patients had these treatments at the time of inclusion for several years and are not always linked to a recent episode. In order to avoid any confusion, each time it was relevant to underline the absence of an active psychiatric episode, we have added in the text “no active psychiatric disorder”.

2. I am confused for the definition of Axis I and II diagnoses in current study. Is Axis I diagnosis based on a psychiatrist's interview? Is Axis II diagnosis based on the 99-item self-report Personality Diagnostic Questionnaire? Axis I and Axis II are clinical diagnoses. I am not sure whether authors used different modality to define those diagnoses. In addition, based on DSM 5 (as authors reported), the concept of axis diagnosis is not applied.

Indeed Axis I diagnosis is based on a structured psychiatrist's interview, the MINI, and axis II diagnosis is based on the 99-item self-report Personality Diagnostic Questionnaire as mentioned in the method section: “The past and current Axis I psychiatric disorders were explored by a psychiatrist using a structured interview, the MINI 5.0.0 [25]. The 99-item self-report Personality Diagnostic Questionnaire 4+ (PDQ-4+) was used to assess personality disorders.”We have chosen to use the PDQ-4+ questionnaire to explore personality disorders because its use is simple and allows the doctor to check the results using a scale of clinical significance. Since the diagnosis of personality disorders is difficult and time-consuming, using this questionnaire with the scale of clinical significance is a good alternative for research.

The interview (MINI version 5.0.0) and the PDQ4 questionnaire are both based on the DSM IV-TR and not on the DSM5. We had mentioned the DSM5 in our article only in the introduction when we pointed out that the presence of symptoms does not always imply the presence of psychiatric disorders. Since this note is true for different versions of the DSM, we have removed the DSM5 precision to avoid confusion: “In this inflammatory context, other patients can develop anxious and depressive symptoms that are not severe enough to meet DSM criteria for depression and anxiety disorders but which correspond to subclinical disorders”.

3. "The psychiatrist also recorded scores from the Montgomery-Asberg Depression Rating Scale (MADRS) [30] and the Hamilton Anxiety Rating scale (HAMA) [31]"..Authors should report the symptom scores at baseline and during the follow-up between cases and controls.

The MADRS and HAMA baseline scores were only presented in Figure 1. We have added the values in Table 1.

As indicated in the "Participants" section, controls do not have a follow-up visit: “The patients were followed for 2 years with a visit every 6 months (overall 5 visits) including both a dermatological and a psychiatric evaluation. Controls had no follow-up”, only patients have multiple visits.

The evolution of patients' MADRS and HAMA scores during the visits has been added as supplementary data (S1 Fig).

4. based on figure 1, we can see several controls rated >= 10 on MADRS and even more cases rated >= 10 on MADRS. The remission criteria of MADRS is commonly defined at 10. Authors said a psychiatrist diagnosed no axis 1 diagnosis in cases and controls, which were conflicting with the rating scores.

Indeed, a MADRS score of less than 10 is most often retained in therapeutic trials to indicate remission of depression. However, the MADRS is not a diagnostic tool for depression that meets DSM criteria, but rather a tool to measure severity and response to treatment. We can also note that clinical studies including patients suffering from depression generally show MADRS scores ≥ 24.

The presence of the disorders was assessed using the Mini International Neuropsychiatric Interview (MINI). This structured diagnostic psychiatric interview has very good psychometric qualities for the detection of major depressive disorders (sensitivity between 86% and 96% and specificity between 79% and 88% with SCID-P, CIDI and expert diagnoses) with an excellent interrater reliability (kappa=1.00) (1998 Sheehan(1)).

The MADRS is a validated tool that has good psychometric qualities. However, its sensitivity and specificity are not 100%. Here are several examples of studies giving the specificity and sensitivity of the MADRS compared to standardized clinical interviews such as the MINI:

 MADRS

Study Population Depression diagnostic cut-off Sensitivity Specificity

2010 Bernstein (2) Psychiatric outpatients MINI 8 97% 33%

 12 92% 59%

2012 Bunevicius (3) Coronary artery disease MINI 10 88% 85%

2014 Kjaergaard (4) Healthy population SCID 10 70% 89%

Therefore the existence of false positives cannot be ruled out and participants who have MADRS scores greater than 10 in our study could be among them. Note that in participants with MADRS scores greater than 10, the most impacted items in the MADRS (inner tension and reduced sleep) are not the items most specifically linked to depression (sadness, suicide, pessimistic ideas, inability to feel).

1. Sheehan DV, Lecrubier Y, Sheehan KH, Amorim P, Janavs J, Weiller E, et al. The Mini-International Neuropsychiatric Interview (M.I.N.I.): the development and validation of a structured diagnostic psychiatric interview for DSM-IV and ICD-10. J Clin Psychiatry. 1998;59 Suppl 20:22-33;quiz 34-57. 

2. Bernstein IH, Rush AJ, Stegman D, Macleod L, Witte B, Trivedi MH. A Comparison of the QIDS-C16, QIDS-SR16, and the MADRS in an Adult Outpatient Clinical Sample. CNS Spectrums. juill 2010;15(7):458‑68. 

3. Bunevicius A, Staniute M, Brozaitiene J, Pommer AM, Pop VJM, Montgomery SA, et al. Evaluation of depressive symptoms in patients with coronary artery disease using the Montgomery Åsberg Depression Rating Scale. International Clinical Psychopharmacology. sept 2012;27(5):249. 

4. Kjærgaard M, Arfwedson Wang CE, Waterloo K, Jorde R. A study of the psychometric properties of the Beck Depression Inventory-II, the Montgomery and Åsberg Depression Rating Scale, and the Hospital Anxiety and Depression Scale in a sample from a healthy population. Scandinavian Journal of Psychology. 2014;55(1):83‑9.

---

## [Editor Report · Decision Letter 3]

8 Feb 2023

Psychiatric symptomatology in skin-restricted lupus patients without axis I psychiatric disorders: a post-hoc analysis

PONE-D-21-34055R3

Dear Dr.
Isabelle Jalenques,

We’re pleased to inform you that your manuscript has been judged scientifically suitable for publication and will be formally accepted for publication once it meets all outstanding technical requirements.

Kind regards,

Nasrin Akter, MPH

Guest Editor

PLOS ONE
---

## [Editor Report · Acceptance letter]

20 Feb 2023

PONE-D-21-34055R3 

Psychiatric symptomatology in skin-restricted lupus patients without axis I psychiatric disorders: a post-hoc analysis 

Dear Dr. Jalenques:

I'm pleased to inform you that your manuscript has been deemed suitable for publication in PLOS ONE. Congratulations! Your manuscript is now with our production department. 

Kind regards, 

on behalf of

Dr. Nasrin Akter 

Guest Editor

PLOS ONE